# O₃ Concentration and Its Relation with BVOC Emissions in a Subtropical Plantation

**Jianhui Bai**

LAGEO, Institute of Atmospheric Physics, Chinese Academy of Sciences (IAP, CAS), Beijing 100029, China; bjh@mail.iap.ac.cn; Tel.: +86-10-829-950-79; Fax: +86-10-829-950-73

**Abstract:** An empirical model of O₃ is developed using the measurements of emissions of biogenic volatile organic compounds (BVOCs), O₃ concentration, global solar radiation, photosynthetically active radiation (PAR) and meteorological variables in a subtropical *Pinus* plantation, China, during 2013–2016. In view of the different structures of isoprene and monoterpenes, two empirical models of O₃ concentration are developed, considering PAR absorption and scattering due to gases, liquids and particles (GLPs), as well as PAR attenuation caused by O₃ and BVOCs. The estimated O₃ is in agreement with the observations, and validation of the O₃ empirical model is conducted. O₃ concentrations are more sensitive to changes in PAR and water vapor than S/Q (horizontal diffuse to global solar radiation) and BVOC emissions. O₃ is positive to changes in isoprene emission at low light and high GLPs, or negative at high light and low GLPs; O₃ is negative to changes in monoterpene emissions. O₃ are positive with the changes of PAR, water vapor and S/Q. It is suggested to control human-induced high BVOC emissions, regulate plant cutting, and reduce NOx and SO₂ emissions more strictly than ever before. There are inverted U-shape interactions between O₃ and its driving factors, and S/Q controls their turning points.

**Keywords:** ozone; biogenic volatile organic compounds; emission; chemical and photochemical reactions; atmospheric substances (GLPs)



## 1. Introduction

Ozone is one of the important greenhouse gases influencing climate change on global scale [1], and a key constituent of pollutants produced through chemical and photochemical reactions (CPRs) in the atmosphere. Emissions of biogenic volatile organic compounds (BVOCs) dominate the global total volatile organic compounds entering the atmosphere [2,3]. BVOCs, sensitive precursors of O₃, are oxidized to produce new gases, liquids and particles (GLPs) in the atmosphere, e.g., blue haze in the forest [4], O₃ and secondary organic aerosol (SOA) [5–10]. In addition, volatile organic compounds (VOCs, including anthropogenic VOCs (AVOCs and BVOCs) and nitrogen oxides (NOx) are the main precursors of ozone [5]. SOA plays a significant role in cloud condensation nuclei formation and cloud formation [11,12]. OH radicals react with almost all atmospheric trace gases and are rapidly recycled. VOCs, OH radicals, NOx, SO₂ and other GLPs react through CPRs and produce new GLPs (O₃, PM₂.₅, SOA, HCHO, etc.). During CPRs, solar ultraviolet and visible energy (UV, VIS) are absorbed and utilized by GLPs through OH radicals and H₂O [13], and OH radicals are important reactants and bridges connecting almost all GLPs in the atmosphere.

There are some limitations and uncertainties associated with OH radicals, O₃, BVOCs and SOA. For example, OH measurement and model simulation [14–16]; the missing and unexplained OH sink/reactivity [16–19]; O₃ simulation [20–23]; BVOC emission measurements and simulations [24,25]; unknown mechanisms of SOA formation from BVOC oxidation [26–28], missing UV energy [13]. Thus, more challenges are still in realizing the atmosphere, biosphere and their interactions [29–32], and better understanding the photochemistry of O₃–BVOCs. In view of the large uncertainties in BVOC measurements

and model simulations [2,24,25], accurate simulations of the total amount of $O_3$ and SOA still have large uncertainties, though the air quality models, such as weather research forecasting coupled with Community Multi-scale Air Quality (WRF-CMAQ) and chemistry model (WRF-Chem), or other models of similar structures and working principles have been used to simulate $O_3$ and SOA and much progress have been achieved. Are there any important driving factors in controlling the changes in the relations in $O_3$–BVOCs and $O_3$–aerosol? More investigations on interactions and mechanisms in $O_3$–BVOCs, $O_3$–aerosol (e.g., SOA) and $O_3$–PAR are necessary, especially under natural atmospheric conditions. It is an urgent need to accurately evaluate the total amount of $O_3$ and SOA contributed from BVOC oxidation. Finally, the interactions and mechanisms in $O_3$ with its driving factors (BVOCs, PAR, aerosols, etc.) under realistic atmospheric conditions can be used as references in future model studies and pollution control of $O_3$.

BVOC emissions, $O_3$, fine particle formations and solar radiation change synchronously, i.e., all kinds of GLPs and solar radiation are multiple processes that interact. The sun provides UV and visible energy and triggers the changes of GLPs (gases, $H_2O$, particles, clouds, haze, etc.) [13]. In view of the challenge in calculating all atmospheric constituent amounts, especially their total PAR absorption and indirect utilization, energy is an important source for all atmospheric constituents and controls their changes, so an energy method is selected. The progress of measurements and an empirical model of BVOC emissions make it possible to develop an empirical model of $O_3$ concentration and study the photochemistry of $O_3$ and BVOCs. The main objective of this study is to investigate the interactions between $O_3$ and its driving factors, e.g., BVOCs, absorbing substances, scattering substances and PAR. Based on the measurements of $O_3$, BVOC emissions and solar radiation, two empirical models of $O_3$ concentration for considering the individual roles of isoprene or monoterpenes are proposed and evaluated. The sensitivity of the $O_3$ changes with its driving factors, and the relationships between the calculated and observed $O_3$ and its driving factors are investigated. Several methods to reduce $O_3$ and fine particle pollution are suggested.

## 2. Instrumentation and Methods

### 2.1. Site Description

$O_3$ concentrations, BVOC emissions and solar radiation were measured at the Qianyanzhou subtropical *Pinus* plantation, Taihe County, Jiangxi province, China (26°44′48″ N, 115°04′13″ E, 110.8 m) from 22 May 2013 to 4 January 2016. In the center of the flux tower, the forest coverage is about 70% within 100 $Km^2$, and the mean forest canopy height is 18 m. Local dominant trees are *Pinus massoniana*, *Pinus elliottii*, *Cunninghamia lanceolata*. The shrubs are *Loropetalum chinense*, *Adinandra millettii*, and *Lyonia compta*. The average slope is 2.8°–13.5°. The annual average temperature is 17.9 °C [33]. The annual global solar irradiance is 4578 MJ m$^{-2}$, and the annual photosynthetically active radiation (PAR) is 7997 mol m$^{-2}$ in 2013 [34]. A 45-m tower was built in this *Pinus* plantation for flux, solar radiation and meteorological measurements.

### 2.2. Methods and Instruments

$O_3$ was measured using an ozone monitor (Model 205, 2B Technologies Inc., Boulder, CO, USA) installed at the same height (23 m) as the relaxed eddy accumulation (REA) and sonic anemometer. The solar radiation measurement system consists of 3 spectral sensors at 270–3200, 400–3200 and 700–3200 nm, respectively, and a recorder with an accuracy of 5% (model TBQ-4-1, 322 Institute of Jinzhou, Jinzhou, China). Global horizontal radiation was observed by the sensor at 270–3200 nm (referred to as Q). Direct normal solar radiation (referred to as D) was measured with a radiometer (270–3200 nm, Model TBS-2, China). Diffuse horizontal radiation (referred to as S) was derived from Q-D × cos(Z), where Z is the solar zenith angle (degree). UV and visible radiation were derived from the difference over 400–3200 nm, 700–3200 nm, from 270–3200 nm, respectively. The solar radiation sensors were placed at the top of a building (10 m above the ground), located

at the Qianyanzhou Experimental Station of Red Soil and Hilly Land, Chinese Academy of Sciences, 800 m away from the flux tower. Temperature and relative humidity were also measured in this *Pinus* plantation using a HOBO weather station (Model H21, Onset Company, Menlo Park, CA, USA) during this campaign. The NOx concentrations were measured at the bottom of the tower using a $NO/NO_2/NOx$ analyzer (Model EC9841, Ecotech Company, Knoxfield, Australia) from 12 June–16 October, 2014.

BVOC emission fluxes were measured using an REA system. It consists of a three-dimensional sonic anemometer, a data logger, and a data acquisition and control unit that were used to collect air samples onto absorbents (Tenax GR and Carbograph 5TD) in stainless steel cartridges [35,36]. The REA system was located at a platform (23 m above ground level). Air samples were collected from 23 July 2014 using a new gradient measurement system, including the same three-dimensional sonic anemometer and a data logger located at the same position during the REA and gradient measurements, and some new pumps (HL-2 sampler, Beijing municipal institute of labor protection, Beijing, China) for collection of air samples at 20 m and 28 m. Air samples were collected in 30 min using the REA and gradient methods. More information about the system of the REA and gradient, cartridge usage, and solar radiation is reported in [34].

The emission fluxes of a given BVOC species ($F_i$) from the REA technique is $F_i = b\sigma_w$ ($C_{up} - C_{down}$), where $\sigma_w$ is the standard deviation of the vertical wind velocity, b is an empirical coefficient, and $C_{up}$ and $C_{down}$ are the concentrations ($\mu g\ m^{-3}$) of the BVOC species in the up- and down-stream cartridges, respectively. The $F_i$ measured using the gradient method is: $F_i = K_{diff} \times (\Delta C/\Delta z)$, where $K_{diff}$ is the eddy diffusion coefficient; $K_{diff} = (k) \times (u^*) \times (z - d)$ for neutral atmospheric stability, k = 0.4 (von Karmons constant); $u^*$ is the friction velocity (m/s); z is the measurement height (geometric value, m); d is the displacement height and assumed to be 2/3 canopy height (m). The mean canopy height (18 m) was used for calculation [34].

Air samples collected using the REA and the gradient method were shipped to a lab at the National Center for Atmospheric Research (NCAR) in Boulder, U.S.A. and in Beijing at the Institute of Atmospheric Physics, Chinese Academy of Sciences (IAP, CAS) for analysis. The procedures for sample analyses have been reported in [34,37–39]. The procedures for sample analyses by gas chromatographs equipped with flame ionization detector at the laboratory (IAP, CAS) in Beijing, China, were similar to those described by Greenberg et al. [36]. A summary of all sampling periods as well as $O_3$ measurements are shown in Table 1.

**Table 1.** Observation periods for $O_3$ and BVOC emission flux measurements. The numbers in parentheses are the number of emission flux measurements.

| Year | Observational Periods | | | | | |
|------|------|------|------|------|------|------|
| 2013 | 22 May–28 May (26) | | 29 June–6 July (29) | | 6 Aug.–13 Aug. (36) | 7 Sep.–11 Sep. (30) |
| 2014 | 18 Jan.–19 Jan. (16) | | 23 July–27 July (9) | | | |
| 2015 | 14 Jan.–19 Jan. (39) | 22 Apr.–30 Apr. (30) | 6 June–16 June (43) | 23 Aug.–4 Sep. (30) | 2 Nov.–7 Nov. (36) | 31 Dec. 2015–4 Jan. 2016 (37) |

### 2.3. Empirical Model of $O_3$ Concentration

Based on the empirical model of BVOC emissions for Qianyanzhou *Pinus* plantation, China [34], and further application of the principle of PAR energy balance, isoprene and monoterpene roles are considered in the development of the $O_3$ empirical model, respectively. Four processes associated with PAR transfer above the canopy are considered:

(1) PAR attenuation due to BVOCs. Isoprene and monoterpene terms represent PAR attenuation by isoprene and monoterpenes, which are assumed to obey the exponential law and described as $e^{-a_1 ISOm}$ and $e^{-a_2 MTm}$, respectively. $a_1$ and $a_2$ are the absorption coefficients of isoprene and monoterpenes (presumed to be united, $mg^{-1}\ m^2$), ISO and MT are the total isoprene and monoterpene emissions in the sampling period, ISO = t × EFI × 0.1, MT = t × EFM, EFI and EFM are isoprene and monoterpene emission fluxes ($mg\ m^{-2}\ h^{-1}$),

0.1 is a normalizing coefficient for isoprene, t (0.5 h) is the sampling period, and m is the optical air mass in the center of the averaging window.

(2) PAR absorption and consumption (photochemical term) due to other GLPs (e.g., NOx, $SO_2$) when they are taking part in CPRs except $O_3$ and isoprene or $O_3$ and monoterpenes, with emphasis on BVOCs through OH radicals and $H_2O$. For example, when isoprene is displayed in the empirical model (Equations (1) and (3)) but monoterpenes are not, the PAR utilization due to monoterpenes is considered in this term and connected through the CPRs with OH radicals. It is similar to monoterpenes when they are considered in the empirical model (Equations (2) and (4)). This term also includes chemical energy converted from PAR absorption by the absorbers. The fraction of global solar radiation absorbed by water vapor in 0.70–2.845 μm is calculated as $\Delta S' = 0.172 \, (mW)^{0.303}$, W (water vapor content in the whole atmospheric column) = 0.021E × 30, and E is the mean water vapor pressure (hPa) at the ground during the sampling period. Assuming only water vapor is considered and the atmosphere is plane-parallel, the solar radiation on the horizontal surface is $I_0 \cos(Z) - \Delta S' = I_0 e^{-kWm} \cos(Z)$, i.e., $e^{-kWm} = 1 - \Delta S'/I_o$, where $I_o = 1.94$ cal min$^{-1}$ cm$^{-2}$ (1367 W m$^{-2}$, solar constant), and k is the mean absorption coefficient of water vapor in 0.70–2.845 μm.

In the visible region (400–700 nm), the important OH radicals in the troposphere are produced from several ways: (a) $NO_2 + hv \rightarrow NO_2^*$, $NO_2^* + H_2O \rightarrow HONO + OH$, which is 50% of that assumed by the traditional $O(^1D) + H_2O$ reaction [40]. The reaction rate reported by Li et al. [40] impacts $O_3$ formation in high NOx emissions significantly [41]. (b) The photodissociation of $CH_3OOH$ [42], (c) the photolysis of $O_3 \cdot H_2O$ clusters [43], (d) the photochemistry of $SO_2$ "on water" [44] and (e) $NO_2$ and $H_2O$ upon photoinitiated by 410 nm light [45]. The "missing OH sinks" in the forest [17,27,46] and related mechanisms and other OH sources should be investigated continuously [47] to improve our deep understanding of the OH photochemistry in the visible region. In view of the above mechanisms, water vapor is used to represent OH radicals, $NO_2^*$, $H_2O$, $SO_2$ and other chemical constituents.

During the CPRs, gases (e.g., BVOCs, $O_3$ and NOx), liquids (e.g., $H_2O$) and particles (mainly BVOC oxidation products, i.e., SOA) react with each other, especially OH radicals, change in concentrations and gas, liquid and particle phases. Considering the concentrations of each GLPs cannot be measured, and many mechanisms associated with $O_3$, BVOCs, SOA formation are unknown at present, the direct PAR absorption and indirect PAR utilization due to GLPs are calculated by the photochemical term using an energy method because water vapor plays an important reactant/bridge in CPRs, especially in the processes of OH radical formation. Thus, water vapor is used as a surrogate to represent all absorbing GLPs, and the photochemical term expresses the entire energy use during the sampling period, associating with all atmospheric constitutes, e.g., NOx, $SO_2$ and other VOCs, when they are not explicitly described in the empirical Equations (1)–(4), apart from the isoprene or monoterpene terms. Whether the photochemical term be representative of the total energy use caused by absorbing GLPs is explained in Section 4.1 when the $O_3$ empirical model considers NOx roles and the UV empirical model considers 2 terms and 3 terms.

(3) $O_3$ absorption in the region (400–700 nm) ($O_3$ term) is described as $e^{-0.1k_3 O_3 tm}$, where $k_3$ is the $O_3$ absorption coefficient in 400–700 nm and $k_3 = 0.053 \times 10^5$ Pa$^{-1}$ cm$^{-1}$; 0.1 is a normalized coefficient. $O_3$ term can be used to study the interactions between $O_3$ and BVOCs in the visible region and is an application from a previous study, the interactions between $O_3$ and NOx in the UV region [48].

(4) The scattering of GLPs (scattering term) is described as $e^{-S/Q}$. Scattering factor S/Q describes the relative amount of GLPs in the atmosphere, including gas molecules, aerosols, clouds, haze and rain. In more detail, when air mass moves from other regions to the study region, the total scattering from the air mass and other GLPs (e.g., NOx, $SO_2$, VOCs, $O_3$ and aerosols) in the atmospheric column during the sampling period (0.5 h) were described using this term objectively.

Isoprene ($C_5H_8$) and monoterpene (($C_5H_8)_2$) have different carbon structures and behaviors [49] and are two dominant BVOCs emitted from plants, contributing about 50% and 15% of the global BVOCs, respectively [50]. They also show different contributions in the formation of $O_3$ and SOA [51–54]. Therefore, it is necessary to study their individual interactions with $O_3$. It should be pointed out that BVOCs, $O_3$, photochemical and scattering terms express the whole atmospheric column amount and their total energy in the whole atmospheric column during the sampling period. In specific, isoprene and monoterpene terms express their individual PAR extinction by the law of exponential attention. Meanwhile, the photochemical and scattering terms describe the total absorbing and scattering energy associated with entire column GLPs, including air mass moves to the study region in the sampling period. The energy distribution in each term is objectively quantified by analyzing the observational data under realistic atmospheric conditions. PAR is an energy source that drives the absorbing and scattering processes, as well as BVOC emissions, no matter how chemical constituents change in the atmosphere. The PAR energy balance at a horizontal plane during the sampling period is described by Equations (1) and (2) for considering the roles of isoprene or monoterpenes, respectively:

$$PAR = A_1{}'e^{-a_1 ISOm} \times \cos(Z) + A_2{}'e^{-kWm} \times \cos(Z) + A_3{}'e^{-0.1k_3 O_3 tm} \times \cos(Z) + A_4{}'e^{-S/Q} + A_0{}' \qquad (1)$$

$$PAR = B_1{}'e^{-a_2 MTm} \times \cos(Z) + B_2{}'e^{-kWm} \times \cos(Z) + B_3{}'e^{-0.1k_3 O_3 tm} \times \cos(Z) + B_4{}'e^{-S/Q} + B_0{}' \qquad (2)$$

The $O_3$ empirical models considering the roles of isoprene or monoterpenes are:

$$e^{-0.1k_3 O_3 tm} \times \cos(Z) = A_1 PAR + A_2 e^{-a_1 ISOm} \times \cos(Z) + A_3\, e^{-kWm} \times \cos(Z) + A_4\, e^{-S/Q} + A_0 \qquad (3)$$

$$e^{-0.1k_3 O_3 tm} \times \cos(Z) = B_1 PAR + B_2 e^{-a_2 MTm} \times \cos(Z) + B_3\, e^{-kWm} \times \cos(Z) + B_4\, e^{-S/Q} + B_0 \qquad (4)$$

where $\cos(Z) = 1/m$. In order to obtain the optimal PAR energy interactions and distributions in the whole atmosphere between $O_3$ and its driving factors that represent most realistic situations and reduce the influences of errors in the measurements of $O_3$, emission flux and solar radiation, strict criteria for observational data were used in the analysis: (1) BVOC emission fluxes (i.e., isoprene and monoterpenes) less than twice the standard deviation of measured values, respectively, and (2) solar zenith angle $< 55°$, (3) S/Q $< 0.5$ (i.e., relative clean atmospheric conditions and low cloudiness).

The development of the empirical model of $O_3$ was based on the measurement and empirical model of BVOC emissions (EMBE) at this subtropical forest [34]. According to the above data selection criteria, 18 and 8 samples were obtained and used in the EMBE models for considering isoprene and monoterpenes, respectively. It means these data are most representative of the ideal interactions between BVOCs and their driving factors under good light and atmospheric conditions (see Section 3.2.1). It also provides a reliable dataset to further study the interactions between $O_3$ and BVOCs. Later, 12 and 11 samples were determined by combining these BVOCs and $O_3$ data for considering the roles of isoprene and monoterpenes, respectively. It still aims to find and determine the basic law in the energy interaction between $O_3$ and its driving factors by using more strict criteria data, similar to that in the EMBE model development [34]. The empirical model of $O_3$ is evaluated in Section 3.1 and 4.3 and modified in Section 4.3. All terms (except PAR) were normalized individually, so as to express their energy equally and objectively [48].

Water vapor as a function of air temperature and relative humidity was calculated using an equation described by Lowe [55], i.e., water vapor can represent air temperature and relative humidity to some extent. In addition, water vapor can represent a dynamic balance between the atmosphere, plants and soil, and the changes of water and water vapor when they take part in CPRs. Therefore, water vapor has more important roles than the air temperature and relative humidity and is used in the empirical model as a proxy of air temperature and relative humidity.

During the sampling period (0.5 h), when air mass associated with the wind speed and direction transports to the study region, the concentrations of BVOCs, $O_3$ and other GLPs (NOx, $SO_2$, etc.), as well as PAR change simultaneously. Their individual roles are described in different terms (i.e., $O_3$, isoprene or monoterpenes, photochemical, scattering

terms and PAR) and considered in model development and later simulation of ozone. It should be noted that all terms in Equations (1)–(4) describe the total column amount of all parameters (e.g., $O_3$, BVOCs, PAR, water vapor and S/Q) in the sampling period. Their representative area can be several decade $km^2$, as the large area representatives of the solar radiation. In a further application of the $O_3$ and BVOC empirical models [34], they describe the $O_3$ and BVOC emissions for the entire forest region from this half-hour to the next half-hour in the daytime, and these empirical models describe the air mass transportation and associated chemical and photochemical processes continuously when the air mass is passing through the study region.

## 3. Main Results

### 3.1. $O_3$ Estimations and Its Validations

The observed $O_3$ concentrations, BVOC emission fluxes measured by the REA technique, solar radiation and meteorological parameters were analyzed to develop the $O_3$ empirical models and determine the coefficients and constants (Equations (3) and (4), samples, n = 12 and 11 for considering isoprene or monoterpenes). The statistical results calculated using Equations (3) and (4), i.e., coefficients and constants, coefficient of determination ($R^2$), average and maximum of the absolute relative bias, $\delta$avg and $\delta$max (%) ($\delta = |y_{cal} - y_{obs}| \times 100/y_{obs}$, $y_{cal}$ and $y_{obs}$ are calculated and observed $O_3$ concentrations), normalized mean square error (NMSE $= \overline{(y_{cal} - y_{obs})^2}/(\overline{y_{cal}} \times \overline{y_{obs}})$ ) [56] and standard deviations of the calculated and observed $O_3$ concentrations ($\sigma_{cal}$ and $\sigma_{obs}$), are shown in Table 2.

**Table 2.** The coefficients and constants, coefficient of determination ($R^2$), average and maximum of the absolute relative bias ($\delta$avg, $\delta_{max}$ (%)), normalized mean square error (NMSE) and standard deviations of calculated and observed fluxes ($\sigma_{cal}$ and $\sigma_{obs}$) for the $O_3$ empirical model considering isoprene and monoterpene roles (situation A and B), respectively.

| Situation | $A_1$ | $A_2$ | $A_3$ | $A_4$ | $A_0$ | $R^2$ | $\delta$avg | $\delta_{max}$ | NMSE | $\sigma_{cal}$ | $\sigma_{obs}$ |
|---|---|---|---|---|---|---|---|---|---|---|---|
| A | −0.038 | −0.562 | 1.976 | 0.226 | −0.164 | 0.996 | 7.4 | 16.5 | 0.007 | 4.64 | 5.97 |
| | $B_1$ | $B_2$ | $B_3$ | $B_4$ | $B_0$ | | | | | | |
| B | −0.049 | −0.026 | 1.399 | 0.219 | −0.186 | 0.993 | 9.1 | 28.0 | 0.013 | 4.65 | 5.71 |

Generally, the simulated $O_3$ concentrations agreed with the observation (Table 2, Figures 1 and 2) for considering the roles of isoprene and monoterpenes. The mean calculated and observed $O_3$ concentrations were 46.85 and 46.88 ppb with a relative bias of 0.07%, and the root mean square error (RMSE) was 3.94 ppb for situation A, and the corresponding values were 38.74 and 38.76 ppb, 0.06% and 4.37 ppb for situation B.

To evaluate the empirical model performance and reduce the simulation error, the air mass used in BVOC and $O_3$ terms in the early morning and late afternoon was doubled to meet a little larger optical air mass in view of the current atmospheric conditions, according to previous simulations of BVOC emissions in some forests. In view of the REA measurements of BVOC emissions are extensively carried out in typical forests in China, e.g., a temperate forest, subtropical *Pinus* and bamboo forests [34], the observed dataset obtained using the REA technique were used for validation, as well as model development. Two methods were used to evaluate the $O_3$ empirical model. Firstly, the mean half-hourly variation was calculated by averaging half-hourly $O_3$ at each time of the day for 22 May 2013–11 Sep 2013 (BVOC emission measurements using the REA technique, n = 104) and is given in Figure 3. The observed $O_3$ concentrations within 2$\sigma$ of the average were selected. The simulated $O_3$ concentrations overestimated the observations by about 30%, and their averages were 53.8 and 41.2 ppb, respectively. It is reasonable considering the uncertainties in BVOC measurements of about 25% [24] as well as $O_3$ and solar radiation measurements. Secondly, the calculated and observed $O_3$ concentrations in single measurements (e.g., in half-hour) in 2013 were also compared, the mean calculated and observed $O_3$ concentrations were 57.7 and 41.8 ppb (n = 104), respectively, i.e., the simulated $O_3$ overestimated the

observed $O_3$ by 36%. The NMSE and RMSE values were 0.1534 and 22.9 ppb. The increased relative bias (i.e., 30% and 36% compared to 0.07% and 0.06%) was caused by using more observed emission fluxes with larger uncertainties in measurements and $O_3$ simulations under all sky conditions (low PAR, high GLP loads, such as clouds and aerosols, see Section 3.2.1, whereas the data used in the development of the empirical model were measured under optimal atmospheric conditions (i.e., high PAR, clean atmosphere and low S/Q) and had smaller uncertainties in observations and simulations. In addition, a large simulation error was caused by the changes in S/Q (GLP loads, including aerosols and clouds) than isoprene and monoterpenes (see Sections 3.2.1 and 3.2.2), especially for high GLP load conditions.

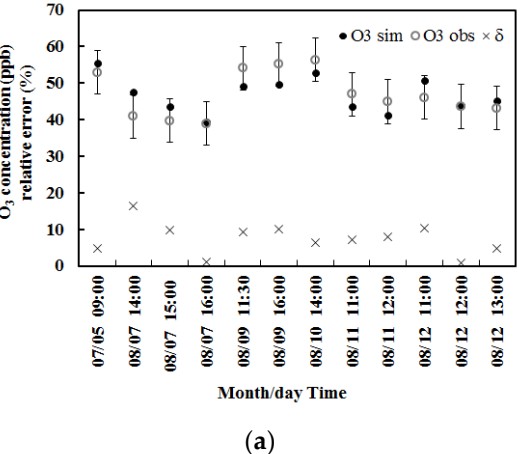

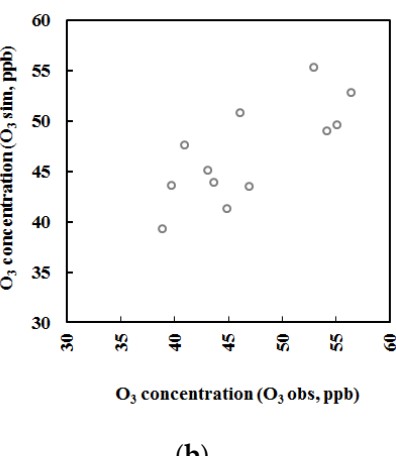

(a)            (b)

**Figure 1.** (**a**) The simulated and observed $O_3$ concentrations ($O_3$ sim and $O_3$ obs), with error bars indicating standard deviation of observed $O_3$ (n = 12) (hereinafter, the same) in a subtropical *Pinus* plantation in China in 2013, and the relative biases (δ, %) (left, situation A, considering isoprene role). (**b**) Scatter plot of $O_3$ concentrations calculated vs. observed (right, situation A).

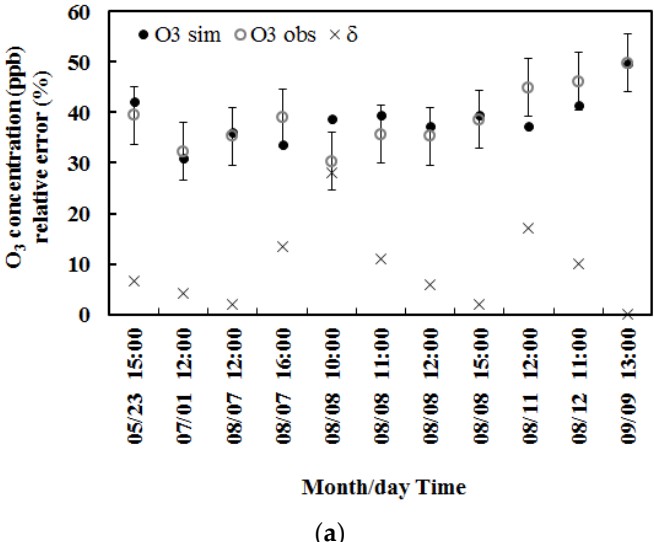

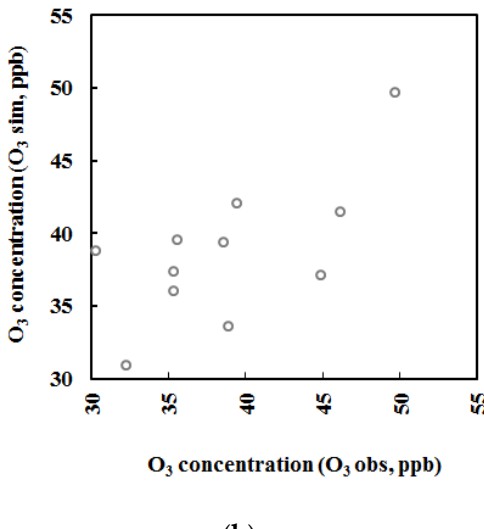

(a)            (b)

**Figure 2.** (**a**) Same as Figure 1a, but for considering monoterpene roles (situation B); (**b**) Same as Figure 1b, but for considering monoterpene roles (situation B).

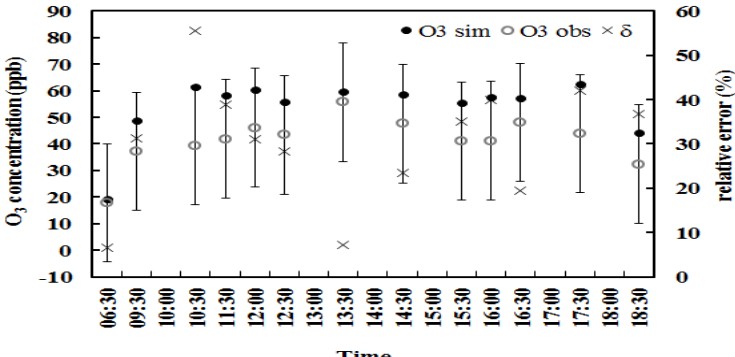

**Figure 3.** The simulated and observed mean half-hourly $O_3$ concentrations with error bars indicating 2 standard deviations of observed $O_3$ (n = 104) in 2013 and the relative biases ($\delta$, %) for situation A.

Similarly, validation for the $O_3$ empirical model considering monoterpene roles was conducted using the observational data in 2013. (1) The mean half-hourly variation was estimated and given in Figure 4 (n = 104). The simulated $O_3$ overestimated the observed $O_3$ (43.8 vs. 41.0 ppb) by 7%. (2) The calculated and observed $O_3$ mean concentrations were 45.3 and 41.6 ppb (n = 104) for single measurements during 2013, respectively, i.e., the simulated $O_3$ overestimated $O_3$ by 9%. The NMSE and RMSE values were 0.1063 and 18.1 ppb, respectively. These uncertainties were reduced compared to that considering the isoprene role, which is mainly due to the monoterpene emissions are the summation of main monoterpene compositions and the random error reduction in the measurements. The comparisons of RMSE values for the $O_3$ empirical model and chemical models are shown in Section 4.3.

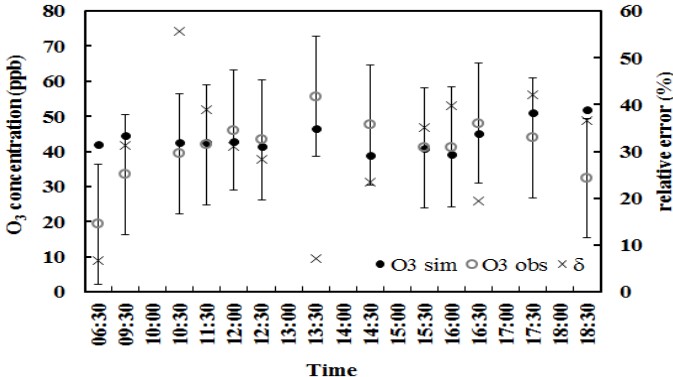

**Figure 4.** Same as Figure 3 but for situation B.

Based on the above validations, the simulated $O_3$ concentrations were in reasonable agreement with those observed, and the empirical models (Equations (3) and (4)) can be used to estimate $O_3$. The representative relationships for $O_3$ and its driving factors were also quantified.

### 3.2. Sensitivity Study of $O_3$ to Its Affecting Factors

To investigate the responses of $O_3$ to its factors under different situations thoroughly, the BVOC emission data (i.e., half-hour measurement) used in the development of the BVOC empirical model (described by REA 12), measured by all REA (described by REA) and all gradient techniques (described by GRA) were selected for the sensitivity tests. These three situations reflect different light and atmospheric conditions (Section 3.2.1).

### 3.2.1. O$_3$ Responses with Its Affecting Factors Considering the Roles of Isoprene

The sensitivity test was performed using a half-hourly observed dataset under realistic atmospheric conditions, i.e., how simulated O$_3$ responses with one driving factor (isoprene, or PAR, E, S/Q) using Equation (3) when the others keep at their original values.

For the above three situations, the average changes in O$_3$ concentrations (%) caused by changes (%) of each factor while keeping all others at their original levels are shown in Figure 5. When extreme changing rates of O$_3$ were removed, sample points were about 12, 111 and 175 for REA 12, REA and GRA, respectively, when considering isoprene role; when considering monoterpene roles, the sample points were 11 (expressed as REA 11), 110 and 150, respectively.

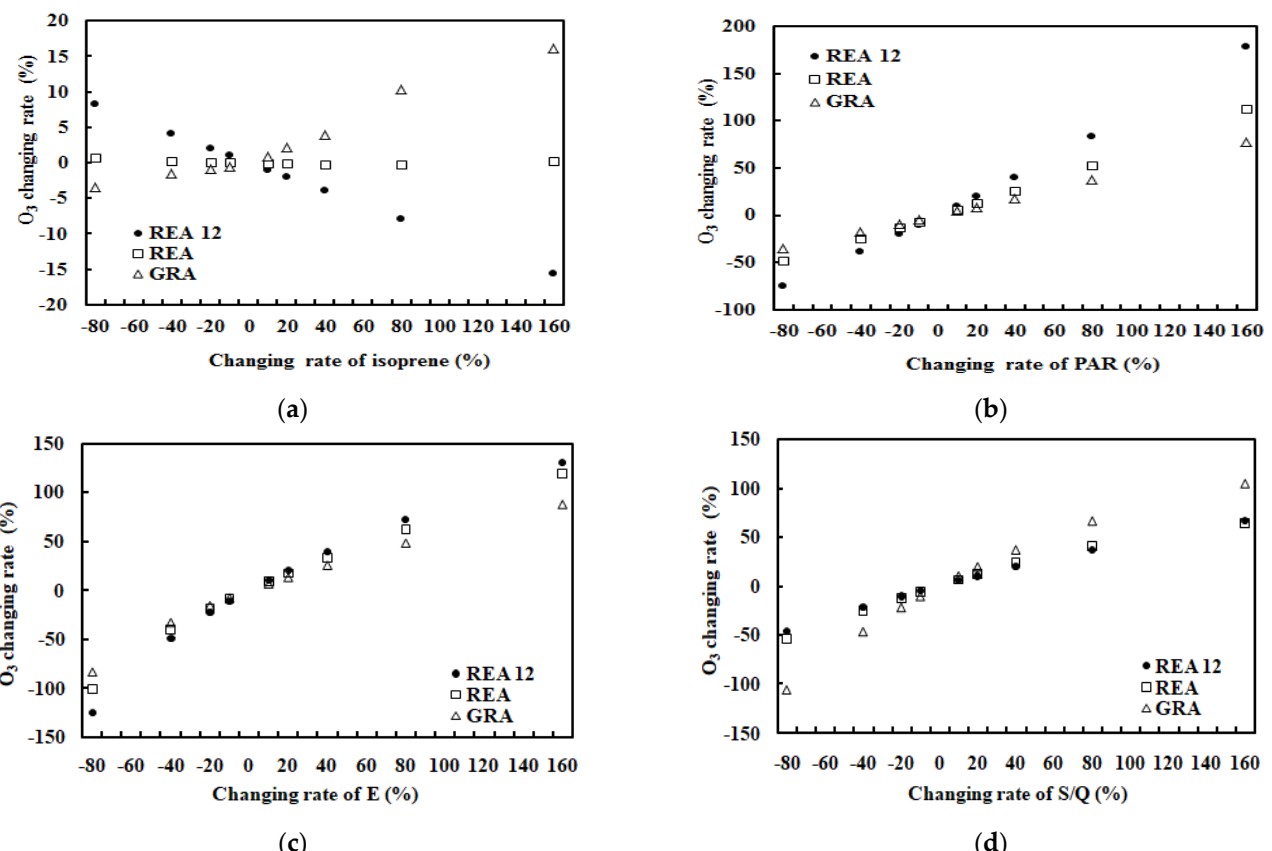

**Figure 5.** O$_3$ changing rates (%) with the change of one factor (isoprene, or PAR, E, S/Q for situations of (**a**–**d**), respectively) and other factors kept at their original levels under realistic atmospheric conditions, REA and GRA denote the emission data measured by REA and gradient technique, respectively. REA12 denotes the isoprene emission data used in model development (O$_3$ empirical model considering isoprene role).

The mean responses of O$_3$ concentration with the changes of each driving factors (isoprene emission, PAR, E and S/Q) by 20% are reported in Table 3. (1) For REA 12, REA and GRA measuring periods, O$_3$ concentrations were more sensitive to E than PAR, S/Q and finally isoprene. E (as well as temperature) and PAR are dominant driving factors and make larger changes in O$_3$ than other factors, reflecting that O$_3$ is strongly produced/destroyed through chemical and photochemical processes triggered by light and temperature, together with water and water vapor supply associated with the OH production. (2) It is well known that O$_3$ formation is contributed by BVOC oxidation at most sites in China [22,23,57–62]. O$_3$ responds (a) negatively with the changes of isoprene (REA12) in good light and atmospheric conditions, i.e., high PAR, high temperature and low S/Q (low GLP loads), and (b) positively with the changes of isoprene (REA, GRA) in bad light and atmospheric conditions, i.e., low PAR, low temperature and high S/Q

(Figure 5, Tables 3 and 4). Similar modeled and observed results, i.e., positive and negative responses, are also reported in [63,64]. Therefore, the positive or negative response of $O_3$ to the isoprene is dependent on light and atmospheric conditions. More and long-time measurements in different forests are necessary to better understand the complex $O_3$–BVOCs interactions. $O_3$ responses to the change of isoprene were larger for GRA than REA and finally REA12, corresponding well to the low, medium and high states of GLPs–PAR, i.e., high GLPs and low PAR (S/Q = 0.75, PAR = 1.26 mol m$^{-2}$), medium GLPs and medium PAR (S/Q = 0.59, PAR = 1.88 mol m$^{-2}$), low GLPs and high PAR (S/Q = 0.35, PAR = 2.75 mol m$^{-2}$). These three situations represent three states: high GLP loads with low air temperature (T) and high relative humidity (RH); medium atmospheric conditions with medium T and RH; clean atmospheric conditions with high T and low RH. Therefore, $O_3$ showed different responses to isoprene, which depended on the state of atmospheric substances and PAR energy.

**Table 3.** $O_3$ changing rates for REA12, REA and GRA (in % and ppb) caused by the changes of one factor at 20%, while other factors kept at their originally simulated levels under realistic atmospheric conditions.

| Situation | ISO | | PAR | | E | | S/Q | |
|---|---|---|---|---|---|---|---|---|
| | % | ppb | % | ppb | % | ppb | % | ppb |
| REA 12 | −2.0 | −0.9 | 20.0 | 9.2 | 20.4 | 9.5 | 10.1 | 4.7 |
| REA | −0.1 | −0.3 | 12.8 | 7.0 | 17.0 | 9.7 | 12.0 | 7.6 |
| GRA | 2.1 | 7.8 | 9.1 | 5.1 | 13.5 | 8.5 | 19.4 | 7.3 |

**Table 4.** Averages of the parameters measured in the establishment of the BVOC empirical model, by all REA and all gradient techniques.

| Measurement | PAR | T | RH | E | S/Q | $O_3$ |
|---|---|---|---|---|---|---|
| REA 12 | 2.75 | 37.2 | 44.1 | 27.9 | 0.35 | 46.9 |
| REA | 1.88 | 32.6 | 63.2 | 30.1 | 0.59 | 40.5 |
| GRA | 1.26 | 21.8 | 76.6 | 22.1 | 0.75 | 39.1 |

It also reveals that $O_3$ is destroyed (−0.9 ppb) by $O_3$ photolysis and reactions with OH radicals and other GLPs with the increase of isoprene when PAR is high at 2.75 mol m$^{-2}$ (1527.8 μmol m$^{-2}$ s$^{-1}$). The chamber experiment shows a similar result; the OH radical initiated photooxidation of isoprene produces methyl vinyl ketone, methacrolein and formaldehyde [65]. $O_3$ depleted a little (−0.3 ppb) with the increase of isoprene emission when PAR is at medium level, 1.26 mol m$^{-2}$ (700 μmol m$^{-2}$ s$^{-1}$). Large $O_3$ is produced (7.8 ppb) with the increase of isoprene emission when PAR is low at 1.26 mol m$^{-2}$ (700 μmol m$^{-2}$ s$^{-1}$). It is clear that PAR = 1.80 mol m$^{-2}$ (1000 μmol m$^{-2}$ s$^{-1}$) or its corresponding state of GLPs–PAR is a controlling/turning point for the positive/negative response of $O_3$ to isoprene (Tables 3 and 4).

The increase of $O_3$ is larger than its decrease when isoprene changes at the same rate (comparing the situation REA 12 to GRA, Table 3), and it is mainly caused by the initial isoprene emissions, 0.39 and -0.61 mg m$^{-2}$ h$^{-1}$. Thus, it is most effective to control isoprene emission at low $O_3$ (39.1 ppb) than at high $O_3$ (46.9 ppb) and control human-induced BVOC emissions, i.e., plant-cutting and biomass burning. For example, plant-cutting (branches and grasses) is suggested to be carried out after 16:00 in big cities, as injured leaves and the grass cutting enhance BVOC emissions dramatically. This mechanism is a reference in future $O_3$ pollution control.

It is interesting that the ratios of REA to GRA for the $O_3$ response to each factor were similar to the ratios of REA to GRA for the average of each factor (Tables 3 and 4), e.g., the above two ratios were −0.1 and −0.2 for isoprene, 1.4 and 1.5 for PAR, 1.3 and 1.4 for E, 0.6 and 0.8 for S/Q, meaning that $O_3$ responses to its driving factor strongly depend on the

mean values of the corresponding factor, and the REA-measured BVOC emission fluxes and gradient techniques have similar features for $O_3$–isoprene photochemistry. Similarly, these features were also found for $O_3$–monoterpenes photochemistry (Tables 4 and 5).

**Table 5.** Same as Table 3 but considering monoterpene (MT) roles.

| Situation | MT | | PAR | | E | | S/Q | |
|-----------|------|------|------|------|------|------|------|------|
| | % | ppb | % | ppb | % | ppb | % | ppb |
| REA 11 | −1.6 | −0.6 | 31.2 | 11.9 | 17.5 | 6.7 | 11.1 | 4.3 |
| REA | 7.7 | 3.5 | 21.1 | 8.7 | 15.2 | 6.5 | 13.6 | 6.1 |
| GRA | 1 | 1.0 | 18.1 | 6.3 | 15.1 | 5.5 | 19.6 | 6.7 |

(3) The mean response of $O_3$ concentration to the changes of PAR was positive, indicating $O_3$ formation is driven by solar energy. (4) The mean response of $O_3$ to the changes of water vapor was positive, revealing that $O_3$ formation is beneficial from sufficient and increased water supply, and more OH radicals are produced from $H_2O$. 5) The mean response of $O_3$ concentration to the changes of S/Q was positive, reflecting fine particles are produced with $O_3$ through CPRs. The $O_3$ responses to all affecting factors were linear, except isoprene.

### 3.2.2. $O_3$ Responses with Its Affecting Factors Considering the Roles of Monoterpenes

Under realistic atmospheric conditions, the $O_3$ responses with each affecting factor (monoterpenes or PAR, E and S/Q) were calculated using Equation (4). The calculating results are given in Figure 6 and Table 5.

For REA 11, REA and GRA measuring campaigns, $O_3$ concentrations were more sensitive to the changes of PAR than E, S/Q and finally, monoterpenes. PAR and E (as well as temperature) are still dominant driving factors and result in larger changes of $O_3$ than other factors. In general, $O_3$ responses with the changes of monoterpenes were negative for three situations, linear for REA11 and REA and non-linear for GRA. A similar modeled result is reported by Nishimura et al. [52]. $O_3$ responses with the changes of monoterpenes were larger for REA11 than REA and smaller for GRA, corresponding well to their clean levels of the atmospheric conditions, which demonstrates that the cleaner of the atmosphere, the more production of $O_3$ contributed from high monoterpene emissions and their oxidation.

For monoterpenes, the mean responses of $O_3$ concentration to the changes of PAR, water vapor and S/Q factors were also positive, indicating that they have similar interactions with $O_3$ as isoprene does with $O_3$. It should be emphasized that the CPRs are homogeneous and heterogeneous, and S/Q can represent the fine particle formation when S/Q and cloud amounts are low.

When monoterpenes increased by 20%, $O_3$ responses (in percent and ppb) with the changes of monoterpenes were larger (REA11 and REA) in good light and clean atmospheric conditions (Figure 6 and Table 5) and smaller (GRA) in low light and polluted atmospheric conditions (Figure 6 and Table 5). The $O_3$ responses with other driving factors (except monoterpenes) were linear and positive.

In general, the responses of $O_3$ to its driving factors are similar when considering the roles of isoprene and monoterpenes and also similar to that in a subtropical bamboo forest in China [66].

Comparing the response of $O_3$ to the scattering substances (expressed by S/Q), the changing rate of $O_3$ was larger for monoterpenes than isoprene. This similar feature is also found in a subtropical bamboo forest [66]. Both indicate that monoterpenes are more productive for the formation of fine particles and $O_3$ than isoprene under the natural atmosphere. Similar results are observed using the chamber experiments, i.e., SOA yields from the oxidation of monoterpenes are larger than isoprene [51,53]. The response of $O_3$ to monoterpenes was larger than isoprene for REA measurements (Figures 5 and 6), and it is

in agreement with the studies that monoterpenes have higher $O_3$ formation potential than isoprene [54,67].

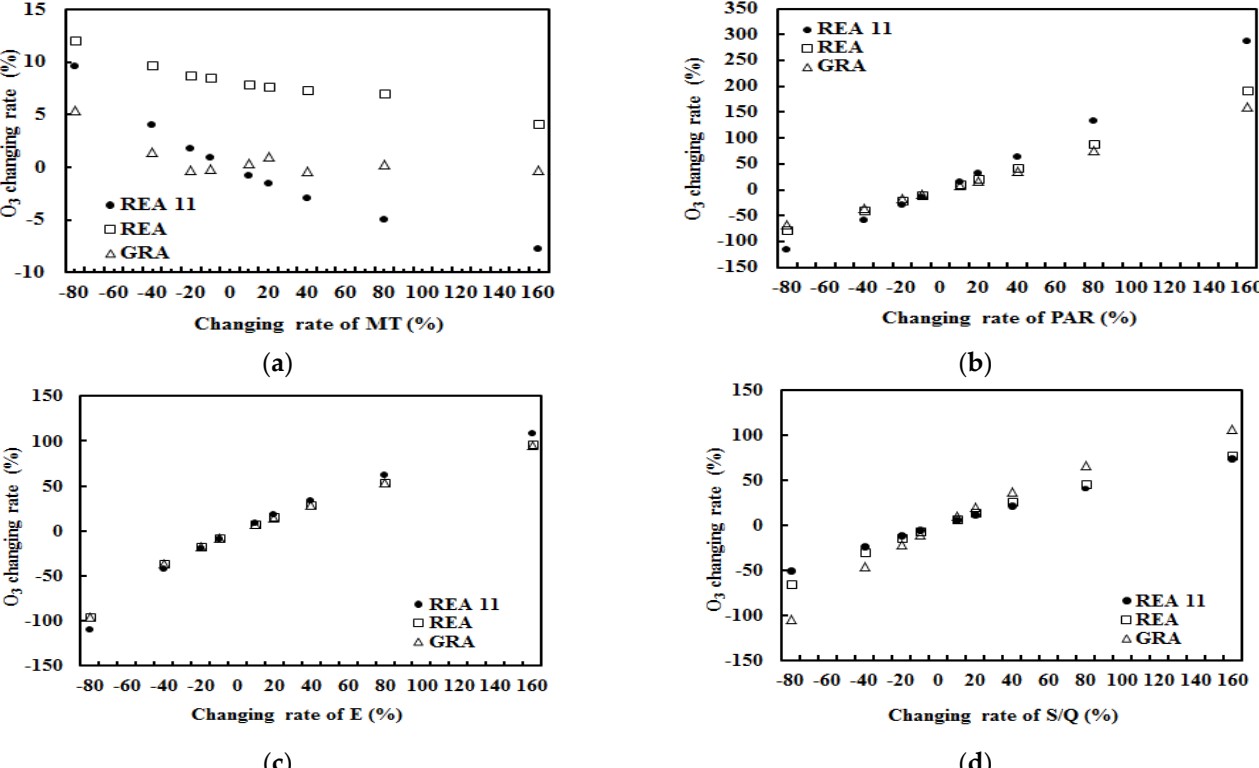

**Figure 6.** $O_3$ changing rates (%) with the change of one factor (monoterpenes, or PAR, E, S/Q for situations of (**a–d**), respectively) and other factors kept at their original levels under realistic atmospheric conditions. REA and GRA denote the emission data were measured by REA and gradient technique, respectively. REA11 denotes the monoterpene emission data used in model development ($O_3$ empirical model considering monoterpene roles).

The linear fitting line between $O_3$ and its affecting factors for the $O_3$ sensitivity test is described by $DO_3 = C1 \times Dfactor_i + C_0$ and shown in Tables 6 and 7. $DO_3$ and $Dfactor_i$ denote the difference of $O_3$ and $factor_i$, respectively. $C_1$ and $C_0$ are the coefficient and constant, respectively. A comparison of the two situations in model development considering the roles of isoprene (REA12) and monoterpenes (REA11) provides the following conclusions: (1) the negative response of $O_3$ to the changes of isoprene was larger than monoterpenes, indicating more $O_3$ is produced/destroyed from isoprene than monoterpenes through CPRs under good light and atmospheric conditions. This characteristic still existed for the GRA situation, with smaller $O_3$ changes under bad light and atmospheric conditions, except the positive response of $O_3$. In contrast, the negative response of $O_3$ to the changes of monoterpenes was larger than isoprene for REA measurement, and the corresponding mean values were PAR= 1.88 mol m$^{-2}$ (1044.4 μmol m$^{-2}$ s$^{-1}$), T = 32.6 °C, RH = 63% and S/Q = 0.59. These light and atmospheric conditions are best suitable for $O_3$ photochemical production and close to the turning point (S/Q = 0.55, Section 4.3), the best interactions between $O_3$ and its influencing factors. It is obvious that the larger sensitivity of $O_3$ to the changes of isoprene or monoterpenes depends on the light and atmospheric conditions (Tables 3–5, Figures 5 and 6). (2) $O_3$ changes with the changes of other driving factors were positive. (3) PAR and E are the primary important factors to make $O_3$ changes.

**Table 6.** The coefficients and $R^2$ of the fitting lines in responses of $O_3$ with its driving factors considering isoprene's role.

| Situation | ISO | | | PAR | | | E | | | S/Q | | |
|---|---|---|---|---|---|---|---|---|---|---|---|---|
| | $C_1$ | $C_0$ | $R^2$ | $C_1$ | $C_0$ | $R^2$ | $C_1$ | $C_0$ | $R^2$ | $C_1$ | $C_0$ | $R^2$ |
| REA12 | −0.989 | 8.970 | 0.999 | 10.54 | −92.53 | 0.966 | 10.06 | −101.2 | 0.951 | 8.486 | −85.82 | 0.939 |
| REA | −0.022 | 0.343 | 0.279 | 6.719 | −59.09 | 0.997 | 8.689 | −85.9 | 0.969 | 4.954 | −47.93 | 0.959 |
| GRA | 0.876 | −6.432 | 0.953 | 4.686 | −41.48 | 0.998 | 6.747 | −67.72 | 0.954 | 4.648 | −44.08 | 0.986 |

**Table 7.** The coefficients and $R^2$ of the fitting lines in responses of $O_3$ with its driving factors considering monoterpene roles.

| Situation | MT | | | PAR | | | E | | | S/Q | | |
|---|---|---|---|---|---|---|---|---|---|---|---|---|
| | $C_1$ | $C_0$ | $R^2$ | $C_1$ | $C_0$ | $R^2$ | $C_1$ | $C_0$ | $R^2$ | $C_1$ | $C_0$ | $R^2$ |
| REA11 | −0.157 | 2.507 | 0.367 | 16.79 | −146.4 | 0.994 | 8.671 | −87.56 | 0.942 | 5.143 | −48.79 | 0.985 |
| REA | −0.295 | 11.30 | 0.933 | 11.26 | −98.57 | 0.995 | 7.480 | −75.93 | 0.943 | 5.907 | −57.85 | 0.965 |
| GRA | −0.690 | 7.241 | 0.905 | 9.515 | −83.74 | 0.997 | 7.480 | −75.93 | 0.943 | 8.563 | −86.00 | 0.946 |

The $O_3$ response rates ($C_1/C_0$) when considering the roles of isoprene or monoterpenes, respectively, are given in Tables 8 and 9. It is more interesting that for three situations, $O_3$ response rates to isoprene were larger than monoterpenes, and $O_3$ response rates to other corresponding factors (PAR or E and S/Q) were the same for the roles of isoprene or monoterpenes, respectively. Therefore, $O_3$ response rates per unit of $O_3$ to its key influencing factors (PAR or E and S/Q) are the same for all kinds of atmospheric conditions. This characteristic may be a useful reference to evaluate $O_3$ formation from BVOC oxidation.

**Table 8.** The $O_3$ response rates with its driving factors considering isoprene's role.

| Situation | ISO | PAR | E | S/Q |
|---|---|---|---|---|
| REA12 | −0.110 | −0.114 | −0.099 | −0.099 |
| REA | −0.064 | −0.114 | −0.101 | −0.103 |
| GRA | −0.136 | −0.113 | −0.100 | −0.105 |

**Table 9.** The $O_3$ response rates with its driving factors considering monoterpene roles.

| Situation | MT | PAR | E | S/Q |
|---|---|---|---|---|
| REA12 | −0.063 | −0.115 | −0.099 | −0.105 |
| REA | −0.026 | −0.114 | −0.099 | −0.102 |
| GRA | −0.095 | −0.114 | −0.099 | −0.100 |

$O_3$ concentrations were most sensitive to the changes in PAR and E (representative of air temperature and relative humidity) than S/Q and isoprene/monoterpenes, implying that total energy (i.e., PAR, sensible and latent heat) is the dominant driving factor in $O_3$ photochemistry (also shown in Section 4.4), then the scattering factor, and finally the BVOCs. BVOCs are recognized as important $O_3$ precursors, but their energy role in PAR utilization is the smallest. The stronger correlations between $O_3$ and its driving factors (PAR, T, RH and E) also show that energy plays a key role in controlling $O_3$ formation and destruction (Section 4.4). Similarly, model results also show that changes in atmospheric conditions dominate the interannual variations of $O_3$ and SOA, and the interannual variations in BVOCs alone lead to small differences (2%–5%) in the calculated $O_3$ and SOA in the summer [68].

## 4. Discussion

### 4.1. $O_3$ Empirical Model

Based on the energy balance between PAR and its transfer processes associated with GLPs in the whole atmosphere, especially considering the PAR attenuations of $O_3$, isoprene

or monoterpenes, their energy interactions and distributions were determined in good light and atmospheric conditions. $O_3$ empirical models considering isoprene and monoterpene roles give reasonable estimates of $O_3$, and their advantages are that the interactions between $O_3$ and its driving factors, PAR, E, isoprene/monoterpenes and atmospheric GLPs (especially fine particles at low S/Q), can be studied extensively.

It is found that the observed PAR and UV have positive correlations with the photochemical term [13], but water does not absorb visible and UV radiation according to previous knowledge (discarding the debate of water absorption in the UV region); thus, the photochemical term represents absorption and indirect use by GLPs through OH radicals during CPRs. More detailed explanations are reported in the UV region [13]. In short, OH radicals are produced in many ways in the visible region (Section 2.3), e.g., $H_2O$ plays an energy use/transfer bridge role in OH radical formation through $NO_2^* + H_2O \rightarrow OH$, implying that $H_2O$ utilizes the visible energy from $NO_2^*$, which is similar to that in other situations (b–e, Section 2.3). Similarly, $H_2O$ utilizes UV energy from $O_3$ in OH radical formation because of a strong positive correlation between UV and the photochemical term [13].

Under S/Q < 0.8 conditions, strong positive correlations were also found between observed monthly UV, VIS, PAR and water vapor pressure at this forest (n = 14, corresponding to 2779 hourly values), their correlation coefficients were 0.953, 0.927 and 0.897, respectively, at the confidence level of 0.001. These results reveal the point of view that UV and VIS utilization is caused by all GLPs through photochemical reactions with OH radicals and $H_2O$, and UV plays more important roles than VIS because of its higher frequency and higher energy.

It is known that OH radicals react with almost all atmospheric GLPs, especially VOCs [69], and visible energy is absorbed and consumed by GLPs in a single GLP phase and gas–particle conversions during CPRs. Many GLPs ($O_3$, $NO_2$, glyoxal, $CH_3CO$ radical, $NO_3$ radical, OClO, CHOCHO, biacetyl, butenedial, BC and other aerosols) are visible radiation absorber [13,70,71]. Others without visible radiation absorption react with OH radicals, and these absorbers, thus, consume visible radiation indirectly. The most important thing is that OH radical recycles quickly. This part energy is expressed by the photochemical term and determined by analyzing observational data and using the multiple-fitting. The photochemical term is an application from a previous study in the UV region [13] to the visible region.

It is necessary to discuss the meaning of the photochemical term displayed in Equations (1)–(4), direct absorption and indirect utilization due to all GLPs in the whole atmospheric column, except $O_3$ and isoprene or $O_3$ and monoterpenes. When a UV empirical model is used to calculate UV at the surface and consider (1) two terms, i.e., photochemical and scattering terms, which are described similar to this study; (2) three terms, photochemical, $O_3$ and scattering terms (as $O_3$ is one important absorber in the UV region); the relative errors of monthly UV are 4.1% for two terms and 3.7% for three terms; the annual mean UV loss (UV at the top of the atmosphere and UV at the ground) caused by photochemical terms using a two-term equation, and $O_3$ and photochemical terms using a three-term equation are 0.57 and 0.54 MJ m$^{-2}$, respectively. These similar results for using two-term and three-term equations indicate that photochemical terms can express the role of $O_3$ when $O_3$ is not explicitly displayed in the two-term equation [13]. This feature was assumed to be existed in the visible region, i.e., photochemical terms described PAR utilization due to all GLPs in the whole atmospheric column except $O_3$ and isoprene in Equation (1), or $O_3$ and monoterpenes in Equation (2), respectively. Therefore, Equations (3) and (4) were used to study the interactions between $O_3$ and isoprene and $O_3$ and monoterpenes, respectively.

One important issue should be mentioned that there is no evident correlation between the absorbing term and scattering term, and their correlation coefficients were 0.368 for using the data of $O_3$ and isoprene in the development of the $O_3$ model (n = 12), and 0.362 for S/G < 0.8 and 0.059 for S/G $\geq$ 0.8 conditions using observed monthly averages during 2013–2016. Therefore, the photochemical and scattering terms are independent and can

be used to describe the PAR absorbing and scattering roles related to the absorbing and scattering GLPs separately.

NOx (NO and $NO_2$) are also important precursors of $O_3$, their roles in $O_3$ photochemistry in a subtropical evergreen broad-leaved forest (Dinghushan, Guangdong province, China) are studied using a similar method and empirical model as this study [48], but the difference is that it is in the UV region and without the consideration of BVOCs. The relative bias and NMSE of hourly ozone concentration are 6.82% and 0.01 for clear sky conditions (n = 113) and 11.30% and 0.02 for all sky conditions [48]. This can be as a reference for the representative of the photochemical term, the total energy absorption and use caused by all of the absorbing GLPs through OH radicals and $H_2O$ in CPRs. In this broad-leaved forest, $O_3$ is more sensitive to its precursors ($NO_2$, NO) than the other factors (UV, E, S/Q), more sensitive to $NO_2$ than NO; the responses of $O_3$ to the changes of driving factors (NO, $NO_2$, UV, E, S/Q) are higher in summer than autumn and higher in clear skies than cloudy skies. If NOx is displayed in the $O_3$–BVOCs empirical model, it would be progressive for understanding $O_3$ photochemistry in the future when NOx is available. If more specific variables and their roles are needed to be simulated and investigated, e.g., BVOC emissions [34], $O_3$ and BVOCs in this study, $O_3$ and NOx [48], these variables can be picked out from the photochemical term and expressed explicitly, and let the roles of the other variables (not described explicitly) described in the photochemical term. When AVOCs are available in the future, their roles to $O_3$ photochemistry can be expressed additionally and studied as a further application of this empirical model. The more variable variables are expressed, the deeper the understanding of $O_3$–BVOCs–aerosols interactions achieved.

BVOC emissions vary with PAR and temperature. BVOCs react with OH radicals and AVOCs and produce new GLPs (e.g., cloud condensation nuclei, SOA, contributing to cloud formation). BVOCs play critical bridge roles to connect the atmospheric substances in gases, liquids and particles to the GLPs and solar energy. More than 30000 BVOCs are released from vegetation [72] and interact with other GLPs and PAR. The interactions between atmospheric substances ($O_3$, NOx, VOCs, $H_2O$, particles, etc.) and light are in many dimensions/directions and non-linear. Numerous atmospheric constituents change in three phases all the time, and many mechanisms associated with CPRs are still not clear, e.g., SOA formation from BVOCs, OH reactivity, but the one important point is that UV and VIS radiation is a very important energy source triggering the CPRs. No matter how the chemical compositions and reactions change, the energy associated with their main processes is a basis to drive their changes in the whole atmosphere. Therefore, the empirical energy method was selected to study the complicated interactions in $O_3$–BVOCs–$H_2O$–other GLPs–PAR. PAR and the energy interactions (Equations (1)–(4)) controlled the changes and interactions of $O_3$, BVOCs and other GLPs (e.g., NOx, AVOCs, and stratospheric $O_3$ through OH radicals). It is an advantage using the energy method because we pay attention to only their energy roles and do not need high or low concentrations of BVOCs and $O_3$, the specific mechanisms discussed in the introduction, how the BVOCs and $O_3$ correlated, etc. The actual roles with the changes of BVOCs and $O_3$ under realistic atmospheric conditions are expressed by their related terms (Equations (1)–(4)). It saves much time in the calculations of the concentrations, chemical and photochemical reaction rates, and PAR utilization of GLPs. A similar $O_3$ empirical model is developed, and similar positive and negative interactions are also found between $O_3$ and BVOCs in a subtropical bamboo forest as in this study [66]. The empirical model of $O_3$ is a specific model for one site currently. It is a beginning using energy method to deal with the complicated $O_3$ and BVOCs photochemistry, and more studies need to be conducted in other forests. Vegetation under low or high accumulated $O_3$ can lead to increased or decreased isoprene emission [20]. This feature in the vegetation is somewhat similar to that in the atmosphere (Section 3.2.1, Tables 3 and 4). Therefore, they should be investigated together for the interaction mechanisms between $O_3$ and isoprene and other BVOCs in the atmosphere and plants.

### 4.2. Implications of Sensitivity Analysis

Equations (3) and (4) represent multi-directional interactions in the natural atmosphere. The positive responses of $O_3$ to the change of S/Q for considering isoprene and monoterpenes indicate that the harmful fine particulate matter (e.g., $PM_{2.5}$, SOA) and $O_3$ can be reduced simultaneously, especially at good light and clean atmospheric conditions. $O_3$ responses to isoprene and monoterpenes are positive in most environmental conditions, indicating that it is critical to control or regulate high $O_3$ formation from BVOC oxidation. Understanding how to regulate high BVOC emissions in cities is essential. It was found that wounded leaves and grass cutting enhanced BVOC emissions dramatically [73,74], and biomass burning results in high BVOC emissions and $O_3$ in this subtropical plantation [34], so it is practical to reduce artificially enhanced BVOC emissions, e.g., changing plant-cutting to after 16:00, reducing biomass burning (e.g., straw) or planting trees and grass with no or low BVOC emissions for large cities so as to reduce high $O_3$ and fine particle formation.

It is realized that $PM_{2.5}$ and $O_3$ are produced by VOCs reacting with OH radicals and other GLPs triggered by UV and VIS [74]:

$$AVOCs + BVOCs + OH + NO_2 + SO_2 + UV + VIS \ldots \rightarrow new\ GLPs\ (O_3, PM \ldots ) \quad (5)$$

China's forests or tree and grass planting areas have been expanding fast. For example, the planting area in Beijing was $26 \times 10^7$ m$^2$ in 2001 and increased by $6 \times 10^7$ m$^2$ compared to 1995. Forests in Hebei province cover 32% of its land territory in 2017 compared to 26% in 2011. The forest area would be $1.14 \times 10^{10}$ m$^2$ and the coverage above 35% in Beijing–Tianjin–Hebei region in 2020. The increasing trees and grasses will enhance BVOC emissions and consume more NOx and $SO_2$, then produce a large number of air pollutants ($O_3$, SOA, etc.). Therefore, stricter control of NOx and $SO_2$ emissions is highly recommended for polluted regions in China, such as the Beijing–Tianjin–Hebei region. In addition, AVOC emissions should also be controlled as suggested [22,74].

Satellite results indicate a greening pattern in 2000–2017 worldwide that is strikingly prominent in China, and China contributes 25% of the global net increase in leaf area [75]. To reach carbon peaking by 2030 and carbon neutrality by 2060, more trees and grasses will need to be planted in China. Considering the current situation and future changes, more stringent measures are much needed in emission control of human-induced BVOCs, AVOCs, NOx and $SO_2$. High techniques to meet higher needs in emission control of AVOCs, NOx and $SO_2$ in the near future are very necessary.

A similar $O_3$ empirical model considering isoprene's role is developed in a bamboo forest as does in this study [66]. The sensitivity shows that $O_3$ responds a) negatively with the changes of isoprene at PAR (1.83 mol m$^{-2}$) and low temperature (T = 13 °C) and (b) positively with the changes of isoprene at PAR (1.97 mol m$^{-2}$) and low temperature (21 °C). It is deduced that $O_3$ responds to isoprene positively at suitable air temperatures (21–32 °C), implying that both increased $O_3$ and isoprene can be achieved by using the potential energy from outside of the $O_3$–isoprene system (i.e., PAR), and negatively at high air temperature (>32 °C) and low air temperature (<13 °C), implying that an energy transport from $O_3$ to isoprene (in $O_3$–isoprene system) through an $O_3$ decrease at states of very high and very low atmospheric system energy (solar global irradiance at the ground together with CpT; Cp is specific heat at constant pressure) when no more extra energy can be used from the outside of the $O_3$–isoprene system. It means that air temperature is an important factor in the photochemistry of $O_3$–isoprene. There were no outputs of simulated $O_3$ considering isoprene or monoterpenes roles when PAR decreased by 100%, indicating that $O_3$ decreased to zero and no more photochemical $O_3$ production during the night (Figures 5 and 6).

### 4.3. Improved Empirical Model of $O_3$ Concentration and Validation

To improve the simulation of $O_3$, the air mass used in the $O_3$ term was multiplied by two(S/Q) for hourly S/Q > 0.7 (2 times S/Q in the development of the $O_3$ model),

considering the increase of the optical path caused by the GLP scattering in high GLP loads. Then, $O_3$ concentrations were calculated using these modified empirical models for considering the roles of isoprene and monoterpenes, respectively. The new validation results are given briefly using the same data in Section 3.1. For considering isoprene's role, the calculated $O_3$ still overestimated the observation, but the relative bias decreased to 23% (51.2 vs. 41.8 ppb), RMSE was 18.9 ppb for single measurements (n = 104) and 15% (47.5 vs. 41.2 ppb) for the half-hourly variation. For considering monoterpene roles (n = 104), the calculated $O_3$ underestimated the observed by 5% (39.8 vs. 41.6 ppb), RMSE was 16.0 ppb for single measurements and 8% (37.9 vs. 41.0 ppb) for the half-hourly variation. Comparing to the previous validation (Section 3.1), the modified empirical models improved the simulations of $O_3$, especially for considering the isoprene situation.

It is necessary to understand chemical model performance over China and East China, i.e., their comparisons between the simulated and observed $O_3$ concentrations. In short, the RMSE values for hourly $O_3$ simulation are 21.6 and 41.0 ppb using the Models-3 Community Multi-scale Air Quality (CMAQ) [76], in the range of 9.4–20.1 ppb using nested air quality prediction modeling system (NAQPMS) [77], and 12.2–62.8 $\mu g\ m^{-3}$ using the Regional Atmospheric Modeling System CMAQ [78]. RMSE values range from 9.9 to 28.1 ppb for daily $O_3$ simulation using the eight regional Eulerian chemical transport models (CTMs) [79], and 10.0 to 32.7 ppb for annual $O_3$ simulation using 14 state-of-the-art chemical transport models (CTMs) [80]. Ye et al. report that the RMSE value is 37 $\mu g\ m^{-3}$ for $O_3$ simulation in 7 days using the weather research forecasting coupled with chemistry model (WRF-Chem) [81]. In general, the performance of the empirical model of $O_3$ is in agreement with these chemical models, though there are differences in time and space scales. With the more reliable BVOC emission fluxes and emission factors available in representative forests in China, model simulations of $O_3$ and SOA would be improved in the future [34,68,82].

During the validation of the $O_3$ empirical model, the observed ambient $O_3$ concentrations, i.e., mixing ratios, were used for considering the individual role of isoprene or monoterpenes. For example, when considering the isoprene role, the empirical model expressed that $O_3$ varies with isoprene explicitly and other chemical constituents (i.e., monoterpenes, NOx, $SO_2$, etc.) non-explicitly in the photochemical term through the OH radicals and $H_2O$ (Section 4.1). It is similar to the situation when considering monoterpene roles.

To further investigate the performance of the energy method and the $O_3$ empirical model, the emission fluxes of isoprene and monoterpenes were considered together as BVOC term to develop a new $O_3$ empirical model (similar to Equation (4)) using the same dataset for considering the monoterpenes (n = 11). The corresponding coefficients and constant, as shown in Table 2, were 0.234, −0.274, 2.080, 1.555 and −1.089. $R^2$ = 0.993, the mean and maximum of the relative bias were 7.43% and 17.87%, respectively. NMSE = 0.007, RMSE values were 3.32 ppb and 8.57%. It can be seen that similar simulations of $O_3$ were also obtained in comparison with considering the roles of isoprene or monoterpenes, respectively (Section 3.1, Table 2). However, the specific roles that isoprene and monoterpenes play were mixed and changed in this new expression. To accurately understand their actual roles, it is better to describe the specific roles of isoprene and monoterpenes explicitly.

### 4.4. Relationships between Ozone and Its Influencing Factors

To understand the relationships between $O_3$ and its influencing factors in natural atmospheric conditions, $O_3$ concentrations during the REA measurements were calculated using the modified empirical models for considering the roles of isoprene (Section 4.3). The atmospheric GLPs (S/Q values) were divided into small groups with the interval of 0.1 between 0 and 1, and the corresponding sample points were 0, 9, 31, 40, 36, 37, 28, 17, and 76 for S/Q and other variables. All other parameters (PAR, air temperature, water vapor pressure, S/Q) were also divided into small groups with the same interval as the S/Q values.

The relationships between the calculated and observed $O_3$ and each of the factors (PAR, water vapor (E), S/Q, air temperature (T), relative humidity (RH)) were non-linear (Figure 7). Generally, the calculated and observed $O_3$ showed similar features and interactions with each of the influencing factors non-linearly, exhibiting that $O_3$ was produced with the increases of PAR, T, RH, E and GLP loads when these factors are at low levels, and then $O_3$ decreased with the increases of PAR, T, RH, E and GLPs after the driving factors reach their peaks. At which, the mean values were about 2.0 mol m$^{-2}$ (=1110 μmol m$^{-2}$ s$^{-1}$) for PAR, 30 °C for T and 62% for RH, and 26 hPa for E, corresponding to a turning point S/Q = 0.5. It is evident that $O_3$ and fine particles are produced simultaneously at low GLP loads (S/Q < 0.5, clean atmosphere), and $O_3$ formation is inhibited at high GLP loads (S/Q > 0.5, polluted atmosphere). It should be emphasized that the production and destruction of $O_3$ and fine particles depend on light and atmospheric conditions. A similar relationship (i.e., Figure 7) between measured $O_3$ and relative humidity and $O_3$ peaks at RH 50%–60% are also found in the Beijing–Tianjin–Hebei region, China, in 2014–2017 [83]. The features shown in Figure 7 are popular or not are needed to be investigated in other regions for better understanding the interactions of $O_3$ and its influencing factors.

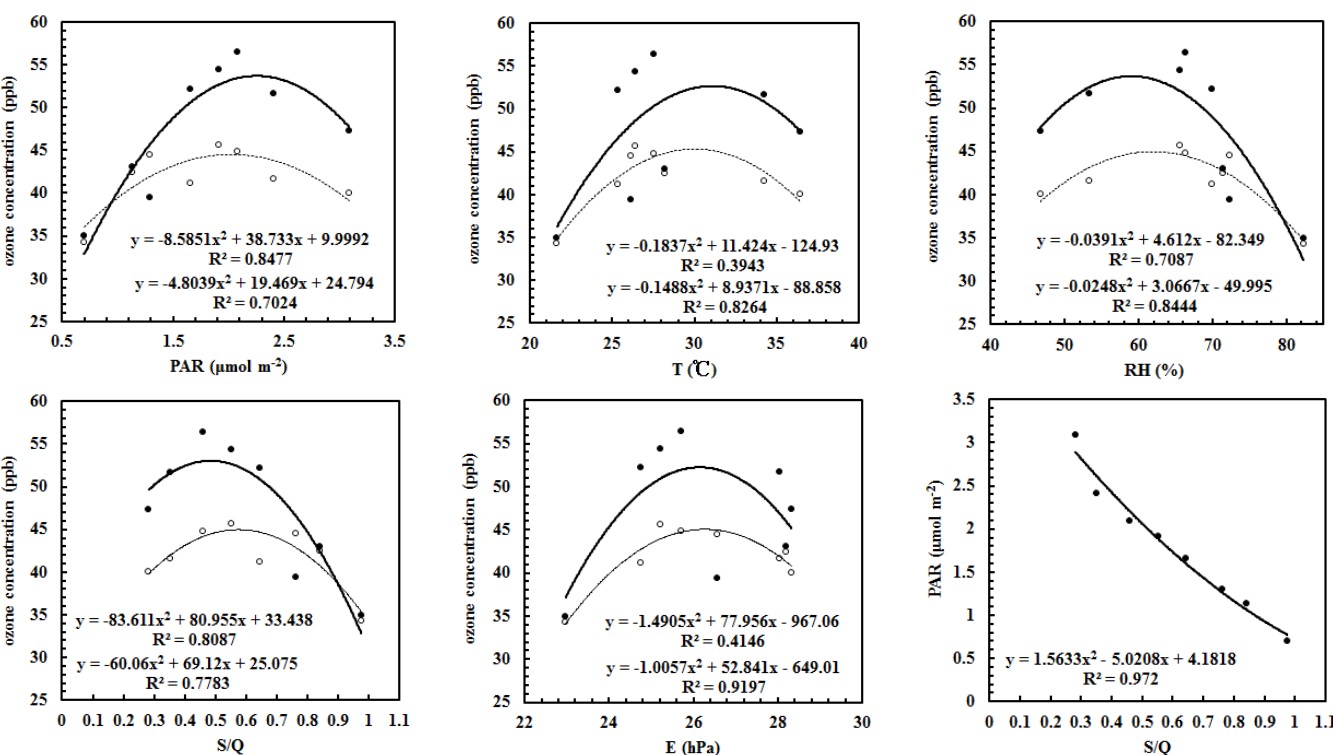

**Figure 7.** The relationships between $O_3$ concentration and its influencing factor, as well as between the observed PAR and S/Q, dark dot and circle denote the calculated and observed $O_3$, respectively. The regression equations were obtained for a polynomial fit.

The relationships between PAR and S/Q were non-linear negatively (Figure 7), reflecting much PAR is attenuated with the increase of GLP loads, including the formation of aerosols, clouds and other GLPs.

The relationships between the estimated BVOC emissions (isoprene + monoterpenes) using the emission model of BVOC emissions [34] and S/Q was inverted U-shape and described by BVOCs = $-13.924 \times (S/Q)^2 + 14.295 \times (S/Q) + 0.3275$ ($R^2 = 0.8813$), and the turning point was also at S/Q = 0.55.

The relationships between hourly NOx concentrations (n = 1367) and S/Q were analyzed using the data measured on 12 June–16 October 2014. NO, NO$_2$ and NOx decreased with the increases of S/Q at low S/Q (<0.75) and increased with the increase of

S/Q at high S/Q (>0.75) (Figure 8). It reflects that the NOx is also important $O_3$ precursors when GLPs are low, but the turning points of NOx (S/Q = 0.75) lag a little compared with other factors, implying an important mechanism that more NOx still participate in the CPRs to produce $O_3$ even at low BVOC emissions, PAR, T and E, after S/Q > 0.55. It is the reason that the stricter reduction of NOx emissions is adopted. After the point of 0.75, no NOx reacted with BVOCs and other GLPs to produce $O_3$, and NOx accumulated in the atmosphere, associating with the decrease of PAR, T, RH and E.

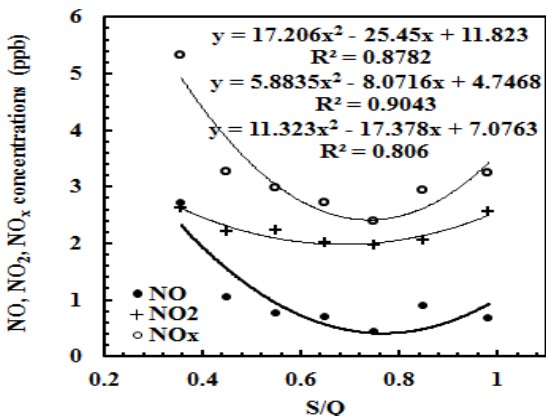

**Figure 8.** The relationships between NO, $NO_2$, NOx and S/Q; the regression parameters obtained from a polynomial fit are for NO, $NO_2$ and NOx from top to bottom, respectively.

$O_3$, BVOCs, water vapor and NOx vary with S/Q non-linearly, and $O_3$, BVOCs and GLPs interact through the light. The GLP amounts are also important factors in controlling the interactions of $O_3$–BVOCs–$H_2O$–GLPs–radiation. The Sun provides the UV and visible radiation to trigger the GLPs taking part in homogeneous and heterogeneous reactions, and UV radiation play more important roles than visible radiation because of its high frequency; thus, much UV radiation is absorbed and utilized by atmospheric GLPs than visible radiation [13]. Water and water vapor are important constituents and sources of OH radicals in the UV and visible regions. Several studies report that organic aerosol (**OA**) contributes more than 50% of the total mass of fine particulate matter (**PM**) during haze events in China, including North China [84–87]; thus, the contributions from VOCs (especially BVOCs) to formation of $O_3$ and fine particles will be more evident and important, in view of the growth of plants in China in the current and future [74,75]. The different S/Q levels represent the approximate equilibrium states of the interactions between the total atmospheric substances and solar radiation energy. The state at S/Q around 0.5 reflects the highest energy state for the atmospheric GLPs, i.e., highest PAR, sensible heat and latent heat, and an optimal interaction between GLPs–light.

To further understand the GLPs–solar radiation system, the changes of GLPs and solar radiation were analyzed when S/Q increased from 0.2 to 0.6: the BVOC emissions and $O_3$ increased (2.6 mg m$^{-2}$ h$^{-1}$ and 6.0 ppb), along with the decreases of NO, $NO_2$ and NOx (2.0, 0.6 and 2.6 ppb), water vapor (3.3 hPa), temperature (9.9 °C), humidity (18.2%), PAR and global solar radiation (632.7 μmol m$^{-2}$ s$^{-1}$ and 239.2 W m$^{-2}$). It is obvious that BVOCs, NOx and water vapor contributed to the $O_3$ and fine particle photochemical formation, alone with much PAR consumption. During new GLP production, the air temperature dropped, which is associated with the decreases of global solar radiation at the surface and the increase in GLPs (T = 25.09 × (S/Q)$^2$ − 48.232 × (S/Q) + 46.907, $R^2$ = 0.7733). When S/Q increased from 0.6 to 1.0, almost all variables decreased, including BVOC emissions (by 2.7 mg m$^{-2}$ h$^{-1}$), water vapor (2.6 hPa), temperature (5.4 °C), PAR and global solar radiation (740.2 μmol m$^{-2}$ s$^{-1}$, 427.3 W m$^{-2}$), $O_3$ (13.0 ppb), NO, $NO_2$ and NOx (0.0, 0.5, 0.5 ppb); however, humidity increased (18.7%). It demonstrates that an increase in GLPs accumulated, which is associated with the increase in NOx, destruction or low production of $O_3$ and low emissions of BVOCs, a great loss of solar radiation in the atmosphere and

the drop of air temperature, implying a mechanism that the accumulation of air pollutants results in a more stable atmosphere. Therefore, BVOCs–other GLPs–solar radiation, i.e., GLPs–light, should be studied as a whole system.

Lee et al. [88] report that UK surface $NO_2$ levels dropped by 42% during the COVID-19 lockdown, but $O_3$ increased compared to previous years, which is attributed to the increased isoprene, UV and temperature. These observed facts provide evidence that the increase of BVOC emissions and UV (together with PAR) result in the production of $O_3$ from BVOC oxidation. These results are in good agreement with the above analyses (Section 3.2 and 4.3). It should be emphasized that the interactions between $O_3$ and its driving factors are very complicated, and the control strategies of $O_3$ should consider the actual states of S/Q (GLPs)–solar radiation, especially in good light and atmospheric conditions and S/Q at low levels. Apart from the driving factors discussed above, other factors, e.g., UV, UV-A, climate change, $CO_2$, warming, bidirectional exchange of BVOCs [89–94], influence BVOC emissions positively or negatively and are suggested to be studied together in the future.

Under UV radiation: the ozone reacts with $H_2O$ to produce OH radical, which reacts with most GLPs in the atmosphere, e.g., BVOCs and AVOCs, NOx and $SO_2$, and then, they produce SOA [13,48,95], and the references therein and benefit cloud formation. Further, it influences the solar UV radiation balance in the atmosphere and on the ground. Similarly, under visible radiation, OH radicals produced from excited $NO_2^*$ react with $H_2O$ and other OH sources [40–44], implying that $H_2O$ utilizes/transfers visible energy from $NO_2^*$ and other GLPs (Section 2.3). In more detail, numerous GLPs absorb visible energy, e.g., glyoxal, $CH_3CO$ Radical, $NO_3$ radical, OClO, CHOCHO, biacetyl, butenedial, NOCl and black carbon [96] and the references therein, and all of these absorbers react with OH radicals and $H_2O$ and transfer absorbed visible energy to other GLPs. Later, absorbers and non-absorbers take part in CPRs, exchange/consume visible energy and contribute to the formation of aerosols and clouds. Visible and UV light play key roles in the formation of SOA and clouds and then air motion but with differences because of their different electromagnetic frequency [97]. Figure 9 shows the mechanism of OH radical production through $H_2O$ and BVOC oxidation in CPRs, aerosols and cloud formation under UV and visible light, as well as the interactions in BVOC emissions and anthropogenic emissions of NOx, $SO_2$, aerosol formation, solar UV and visible light. These interactions are in multiple directions and occur simultaneously in the natural atmosphere. Given the numerous limitations in measurements and simulations, together with large uncertainties and unknown mechanisms in OH [98], SOA production from BVOC's reaction with $O_3$, $NO_3$ and OH + NOx in aqueous photochemistry [99] and from isoprene oxidation [100], missing OH sinks and unmeasured VOCs [46,96], the uncertainties in the application of the laboratory results (e.g., chamber) under controlled conditions to the natural atmosphere and thousands of BVOC compounds and their heterogeneous CPRs, it is impossible and a challenge to measure and simulate each chemical constituent and chemical and photochemical feature (Section 1) [28,72,73,96–100]. Therefore, it is an optional and practical method to study so complicated interactions and systems by grasping the key absorbing and scattering energy processes in the atmosphere. The rationality of this point of view is well proven by clear evidence that PAR energy drives the changes of all chemical compositions as well as all GLPs (e.g., $O_3$, water vapor, S/Q, NOx, Figures 7 and 8) and some results from $O_3$ empirical models were in agreement with the studies of the chamber and chemical models.

In terrestrial vegetation regions, the enhancements of tropospheric column HCHO concentration is consistent with BVOC responses [100], and increased HCHO vertical column density (VCD), surface $O_3$ and aerosol optical density (AOD) are contributed by BVOC oxidation [74], revealing that BVOCs play critical roles in the formation of HCHO, $O_3$ and aerosols in a realistic atmosphere. According to the above discussion, it is better to combine the chemical models focusing on the specific compositions and empirical models

focusing on energy use and distribution to study the mechanisms of $O_3$–BVOCs–aerosols–solar radiation thoroughly.

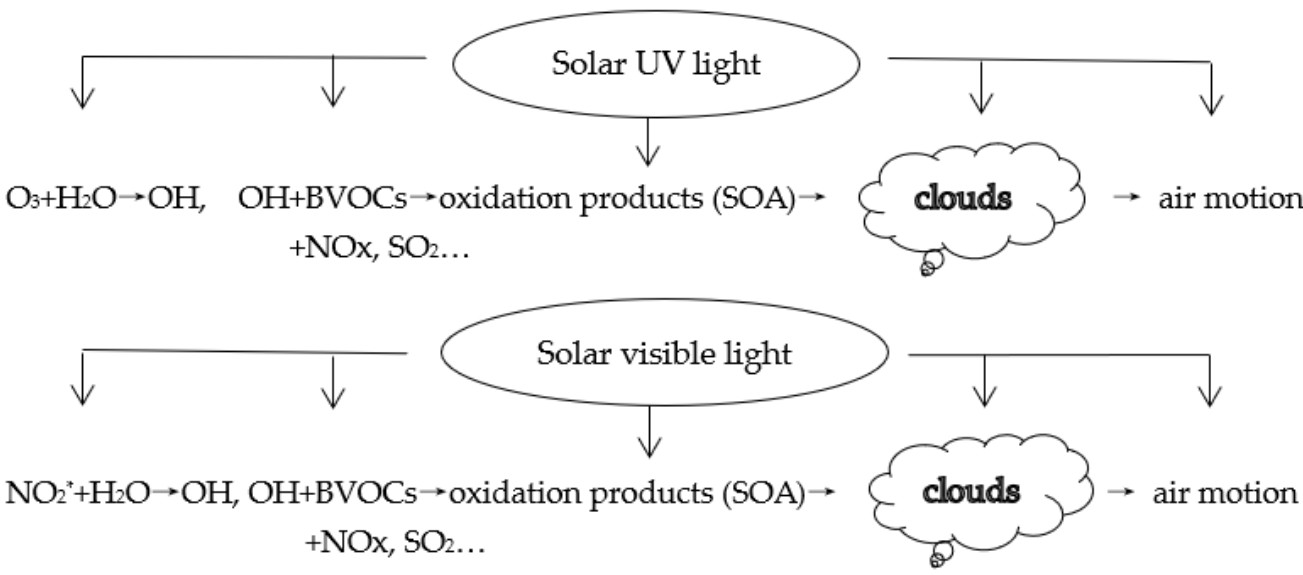

**Figure 9.** Under solar UV and visible light, OH radical production and chemical and photochemical reactions between OH, BVOCs, SOA and $O_3$ in gas, liquid and particle phases in the atmosphere. SOA contributes to clouds formation, and then clouds attenuate solar UV, visible and near-infrared radiation and influence the regional radiation balance and air motion. The emissions of BVOCs and their oxidation, along with the above associated potential effects, are interacted with and controlled by UV and visible energy.

In a short summary, the $O_3$ empirical model show similar performance to chemical models, and several similar results, e.g., $O_3$ and SOA formation, $O_3$ response to its driving factors that obtained in the laboratory and using different chemical models. There is also agreement in the interactions between calculated and observed $O_3$ and its driving factors. All of the above results indicate that the empirical model of $O_3$ can be used to study the photochemical mechanisms of $O_3$ and BVOCs. The energy method has a unique advantage, but the empirical model based on only the correlation between pure numbers may do not have. Visible and UV radiation provides an important energy source to the GLPs in the photochemical reactions in the whole atmosphere and is a critical connection to grasp and understand $O_3$–BVOCs–aerosols–radiation interactions.

## 5. Conclusions

Based on the principle of PAR energy balance, the empirical models of $O_3$ concentration for considering the roles of isoprene or monoterpenes were developed for a subtropical coniferous forest. The calculated $O_3$ concentrations were in agreement with those observed with relative biases of 7.4% and 9.1% for considering isoprene and monoterpenes, respectively. Reasonable validation results of the $O_3$ empirical models were obtained. $O_3$ concentrations were more sensitive to the changes in PAR and E than S/Q and finally isoprene or monoterpenes. It implies that $O_3$ is produced through CPRs. Most responses of $O_3$ with its affecting factors were positively linear, except monoterpenes. The responses of $O_3$ to isoprene were negative at high light and low atmospheric GLP loads or positive at low light and high atmospheric GLP loads. The air temperature also played a key role in the negative or positive responses. Moreover, the responses of $O_3$ to monoterpenes were negative. Monoterpenes are easier to oxidize to fine particles than isoprene. The positive response of $O_3$ to the change of S/Q indicates that $O_3$ and fine particles are produced synchronously, and artificially enhanced BVOC emissions should be controlled.

In natural environmental conditions, the relationships between $O_3$ and its driving factors were non-linear. $O_3$ increased with the increase of PAR, temperature, humidity,

water vapor and GLPs simultaneously during low GLP loads (S/Q < 0.55). S/Q at 0.55 is a turning point between positive and negative interactions for most variables (BVOCs, $O_3$, T, RH, E, NOx and PAR). It is the highest atmospheric substances-energy state, i.e., the sufficient and optimal interaction in GLPs ($O_3$–BVOCs–$H_2O$)–light, representing the best conditions for $O_3$ photochemistry and controlling the directions of the GLP changes. After this point, $O_3$ decreased with the increase of PAR, temperature, humidity, water vapor and GLPs.

It is suggested to regulate human-induced BVOC emissions, and adopt a more stringent reduction standard in AVOCs, NOx and $SO_2$ emissions, so as to reduce $O_3$ and fine particle formation. The energy method deserves to be studied for a better understanding of the $O_3$–BVOCs–$H_2O$–GLPs–radiation system and interactions between the atmospheric substances and solar radiation.

**Funding:** This research was supported by the National Natural Science Foundation of China (grant no. 41275137), Dragon 4 and 5 projects (ID 32771 and 59013) and the European Union (EU) 7 framework programme MarcoPolo (grant no. 606953).

**Institutional Review Board Statement:** Not applicable.

**Informed Consent Statement:** Not applicable.

**Data Availability Statement:** Not applicable.

**Acknowledgments:** Data used in this research measured at Qianyanzhou Station are published on a Big Earth Data Platform for Three Poles, http://poles.tpdc.ac.cn/zh-hans/ (accessed on 13 March 2021). The Relaxed Eddy Accumulation system was provided by the National Center for Atmospheric Research, which is sponsored by the US National Science Foundation. The author thanks all the people for their great assistance, including Alex Guenther at the University of California, Andrew Turnipseed at 2B Technologies, Inc. Boulder, CO 80301, USA, James Greenberg and Tiffany Duhl at the National Center for Atmospheric Research, Boulder, CO 80307, USA; H.M., Wang, F.T., Yang, Q.K., Li, G.Z., Liu, L. Huang, Y.G., Wang, S.Y., Yin, J.D., Zou, J.Z., Zhang, Y.F., Huang, G.L. Zhu at Taihe County, Jiangxi province, and X.W. Wan and Y.M. Wu from the Institute of Atmospheric Physics, CAS. The author thanks the Qianyanzhou Experimental Station of Red Soil and Hilly Land (CAS) for providing meteorological and solar radiation data from January to May 2013. The author also thanks all reviewers for their constructive comments and beneficial suggestions.

**Conflicts of Interest:** The authors declare no conflict of interest.

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
