# Peer review of "O3 Concentration and Its Relation with BVOC Emissions in a Subtropical Plantation"

_atmosphere, doi:10.3390/atmos12060711_

Round 1

Reviewer 1 Report

This paper develops an empirical relationship for predicting O3 concentrations in a measurement site dominated by biogenic emissions based on radiation measurements and a number of adjustable parameters. Unfortunately, the "model" they use and discuss in Section 2.3 is either very poorly explained or is not scientifically sound, either of which would make the paper unacceptable for publication.

Given below is my interpretation of the model they use and the assumptions behind it, why I think they are not consistent with our current understanding of the atmospheric chemistry of ground-level ozone formation. The model is based on the assumption that only the following factors affect O3 concentrations in this biogenic emissions-dominated environment:

1) The emissions fluxes of biogenic species. It is reasonable to assume that this will affect O3 levels because their gas-phase reactions in the presence of NOx will cause O3 to increase, while their direct reactions with O3 (with which most BVOCs react) might cause O3 to decline (though not always). Note, however, that this does not affect O3 formed downwind and transported into the sampling area. This is despite the fact that it looks like they are attempting to estimate local O3 concentrations, not O3 formation fluxes. Their model has no factors that account for O3 transported from downwind.

2 & 3) Light absorption by the biogenic VOCs and by water. Light absorption by isoprene, terpenes, and water is negligible at wavelengths that occur in the lower atmosphere, as is obvious from their well-known UV-visible spectra. If these terms in their model have any predictive effect, it must be totally coincidental or due to them being surrogates for some other factors that were not considered.

4) A "scattering term" that depends on the ratio of the diffuse to the global horizontal radiation. I don't understand why they think this should be a term in their model, and there was no attempt to explain it that I could find.

Not included in the model are factors other than light intensity that I would think would be much more important, which include (in expected order of importance according to my estimations): (1) Wind direction and speed if the sampling site is anywhere other pollution sources (not discussed in the paper); (2) NOx levels (which affects rates of O3 formation, including whether O3 is net formed or destroyed after it reacts with BVOCs); (3) temperature, which affects rates of reactions. There may well be other factors that may be useful in such a empirical model, but these are the most obvious ones that obviously should be looked at first. Note that they have data to allow use of any of these factors in their model.

Of course, it is possible that their model may indeed be the best way to fit the O3 data at their site with the least number of parameters, which may be an interesting result that could be worth publishing. However, there was no apparent attempt to look at other parameterizations or to show that what they came up with performs better than a parameterization that is more consistent with our understanding of the actual processes that may be occurring.

If my interpretation is incorrect and their model is actually scientifically sound, then the authors need to re-write at least Section 2.3 and re-submit this for consideration. If my interpretation is correct but the authors believe that their model is still useful as a purely empirical relationship, then the author needs to remove the incorrect or misleading statements from Section 2.3 and provide data and results that show that their parameterization is at least as good as more chemically reasonable or at least simpler alternatives.

Reviewer 2 Report

The paper investigates factors influencing ozone concentration in subtropical plantation in China using the advanced empirical model elaborated by the author. It is generally well done, quite interesting and novel, and worth to be published after some minor redaction.

General remarks

  1. SO2 is several times mentioned in the paper (also in abstract and in important conclusions) but it is not under consideration in the study.
  2. It is not quite clear what is meant as “human-induced BVOC”. BVOCs produced by industry (like isoprene) are just small part of total BVOC emissions. Does it mean BVOC emissions from artificial forests? This needs to be explained.
  3. In some places text is too complicated and difficult for perception. I would recommend additional proofreading to improve this as well as to check additionally for grammar.

Other remarks

Table 1. C – should be placed in 1 line; F – why is 9 used 3 times only here?

I did not understand why observation periods are marked by letters if they are not used elsewhere.

There is also a contradiction in the table 1 title (Information for emission measurements, while “Numbers in parentheses are the number of flux measurements”. I would rename the table 1 as “Observation periods ….”

Line 181-182 – equations are shifted

Fig. 1. Figures are shifted. I would replace O3 cal by O3 sim (simulated)

It is better to insert whitespace in indices (e.g. O3 obs instead of O3obs)

The same for Fig. 2 and after

Line 248 – the dot after Sep should be removed

Line 263 – should be capital T in 1) The

Line 278 It looks like word “more” is missed before “important”

Line 293. Ratios or rates as on the plot?

Line 305. This conclusion is vague. Does O3 response to any change of isoprene equally under curtain atmospheric conditions? Here and later – what is “good atmospheric conditions”? Such term does not sound scientifically and should be substituted.

Line 334 Should be “burning”

Line 335. I couldn’t get the idea of this passage. How can one govern isoprene emissions based on ozone concentration?

Tables 3 and 4 are shifted

Line 557 – the equation is shifted

Line 576 “To increase” probably should be replaced with “to improve”

Fig. 7 and its discussion – Taking into account rather small amount of data for polynomial approximation it would be reasonable to compare obtained fits with results of other similar works

Lines 717-719 One of conclusions is “the responses of O3 to isoprene were negative at good light and atmospheric conditions, and negative at bad light and atmospheric conditions” while in abstract it is stated that “O3 is positive or negative to changes in isoprene emissions”. This must be clarified.  How can it be explained that the dependence is always negative? What is going on at night in this case?

Reviewer 3 Report

The authors describe the work of establishing an empirical model for O3 formation owing to isoprene and monoterpene emissions with measurements data collected in a temperate forest in China. The technique they used is based on relaxed eddy accumulation (REA) with observations fixed at a height of 23 m for flux measurements of these two BVOCs. Two PAR empirical equations designated for isoprene and monoterpene are established for the use of O3 empirical models attributed to the emissions of these two BVOCs. To test the performance of the O3 model, comparison was made by comparing the observed O3 with the calculated O3 from the empirical models. This type of work is useful and in many cases, important. For instance, the empirical models if properly established can be put in numerical models to better simulated O3 formation due to BVOC in a large part of the Asian continent where BVOC could account for a significant fraction of O3 formation, and the air quality in China and other developing regions in Asia are seriously plagued by high O3 problem. As a result, the authors could further stress the significance and potential implication of their work in helping air quality numerical models in this regard. Overall, the paper is of sufficient merit and quality to be accepted for publication if their writing can be further improved to make reading less painful and more pleasant. A few comments are made as follows.

  1. O3 and SOA formation are mostly done by air quality models such as WRF-CMAQ and WRF-Chem, or other models of similar structures and working principles. Are these empirical equations depicted in this study intended to be used in these types of numerical models as modules of BVOCs? If yes, state the deficiencies of current practices, methods, approaches, etc. and the merits that are going to be gained with the inclusion of their empirical equations. This argument can be placed in the introduction section for readers to get a bigger picture of the usefulness of their work. Who are the stakeholders to benefit from their findings?

  1. 1-4. When validating the models, O3 observations are compared with the simulated data. The problem is that O3 has a background contribution, and how can ambient O3 be compared with the calculated O3 which is independently calculated either by the source function of isoprene or monoterpene? Furthermore, the obs. O3 can come from local formation or transport from elsewhere. They should state with clarity in the text the meaning of obs. O3. Is it ambient O3 mixing ratio or not? If yes, then a direct comparison does not make a whole lot of sense.    
  2. Both isoprene and monoterpene were measured by absorption cartridges which is a rather slow sampling method. How can the REA method couple with a slow sampling technique, e.g, a few minutes to hours, to deduce flux? How many cartridges were collected? While other parameters such as solar radiation, temperature and humidity, etc. are relatively easy to measure by direct reading devices with fairly small uncertainty, the measurements of the two BVOCs are entirely different thing by contrast. They cannot be instantly measured like solar radiation. Therefore, averaging of solar ration and other non-chemical parameters is unavoidable to match the time-scale of BVOC measurements. State more clearly of how this data averaging is performed. How many BVOC samples were collected. In these figures, is n= 11, 12, or 104 the number of cartridges? Why is there a large difference in N, say, 11 vs. 104?
  3. The uncertainty of BVOC has to be much greater than the data of solar, temperature measurements, and yet the agreement between O3 obs. and O3 calculated is only as small as 0.06 or 0.07% in Fig. 2, which is too good to believe considering the uncertainty of cartridge measurements. Yet, in Fig. 3 the difference is as large as 36%. The authors attribute this large error to the uncertainty of BVOC measurements, if so then why this factor did not affect Figs 1 and 2?
  4. On page 6, second paragraph, “To evaluate …in early morning and late afternoon…”. Why reducing simulation error has anything to do with early morning and later afternoon data? No explanation is given. Readers have to scratch their heads without knowing the reason.
  5. On page 7, last paragraph, the authors self-claim the reasonable agreement with the obs. despite limited data points and then said “data quality is important than data quantity.” Limited data is a major weakness and should not be justified by claiming their “good” data quality. Is 36% reasonably good?
  6. Why is O3 obs. independently compared with isoprene or monoterpene induced O3 and not with the combined production of O3? The logical thinking is that O3 obs. is produced from all VOCs combined and therefore the comparison should also be alike. The rationale is missing in the text.

Round 2

Reviewer 1 Report

I am sorry but I am unable to understand how the explanation provided addressed my comments and difficulty in understanding the scientific basis of their model. The new discussion only increased my confusion about the thinking about the model, and also increased my suspicion that the author may not adequately understand the processes of ozone formation and transport in the lower atmosphere. Reference is made to light absorption by water and VOCs even though they do not actually absorb light in the lower atmosphere, and the revision does not clarify this. It is stated in several places that water is used as a "surrogate" for OH, NO2. H2O, other chemical compounds and temperature, but no reason is given as to why we would expect a correlation between them. It is stated that various terms in their model account for transport, but I don't see how. Overall, I have the same criticisms of this version than the previous one, only maybe more so.

Therefore, my recommendation not to publish this paper is unchanged.

Reviewer 3 Report

The English can be improved further especially the later-added text in the revised MS. In fact, the whole MS can be proofread by a professional editing service to improve the readability. 
